

# The effect of biochar amendments on phenanthrene sorption, desorption and mineralisation in different soils

Eduardo Moreno Jiménez[1], Sara Aceña-Heras[1], Vladimír Frišták[2], Stefanie Heinze[3] and Bernd Marschner[3]

[1] Department of Agricultural and Food Chemistry, Faculty of Sciences, Universidad Autónoma de Madrid, Madrid, Spain

[2] Department of Chemistry, University of Trnava, Trnava, Slovak Republic

[3] Department Soil Science/Soil Ecology, Institute of Geography, Ruhr-Universität Bochum, Bochum, Germany

## ABSTRACT

The contamination of soils and waters with organic pollutants, such as polycyclic aromatic hydrocarbons (PAHs), affect a large number of sites worldwide that need remediation. In this context soils amendments can be used to immobilise PAHs while maintaining soil functioning, with biochar being a promising amendment. In this experiment, phenantrene (Phe) was used as a frequent PAH contaminating soils and we studied the effect of three biochars at 1% applications to three different substrates, two agricultural topsoils and pure sand. We evaluated the changes in soil properties, sorption-desorption of Phe, and mineralisation of Phe in all treatments. Phe in pure sand was effectively sorbed to olive pruning (OBC) and rice husk (RBC) biochars, but pine biochar (PBC) was not as effective. In the soils, OBC and RBC only increased sorption of Phe in the silty soil. Desorption was affected by biochar application, RBC and OBC decreased water soluble Phe independently of the soil, which may be useful in preventing leaching of Phe into natural waters. Contrastingly, OBC and RBC slightly decreased the mineralisation of Phe in the soils, thus indicating lower bioavailability of the contaminant. Overall, biochar effects in the two tested soils were low, most likely due to the rather high soil organic C (SOC) contents of 2.2 and 2.8% with Koc values in the same range as those of the biochars. However, OBC and RBC additions can substantially increase adsorption of Phe in soils poor in SOC.

# INTRODUCTION

Petroleum hydrocarbon contamination is one of the most common problems in soils and sediments (*Gerhardt et al., 2009*) and includes a wide range of alkanes, cycloalkanes, and aromatics with a significant level of polycyclic aromatic hydrocarbons (PAHs). PAHs are an environmental concern and represent persistent structures with potentially hazardous effects on environmental and human health in terms of their toxicity, mutagenicity and carcinogenicity (*Tsai et al., 2001*). Phenanthrene is a widespread three-ringed PAH which is rather persistent and difficult to remove from the environment (*Tang et al., 2015*).

Corresponding author
Eduardo Moreno Jiménez,
eduardo.moreno@uam.es

Sorption is a crucial process controlling the mobility, availability and fate of organic contaminants in soil and aquatic systems. The sorption behaviour of PAHs is mainly regulated by the content of soil organic matter (OM), a parameter that can easily rise in soils by amending with carbonaceous products. During the last decades many efforts have been made to explore and produce new materials as an efficient, low-cost soil remediation tool and the application of biochars in soils, mainly produced from wastes, has been intensively studied (*Beesley et al., 2011*; *Gómez-Eyles et al., 2013*; *Ahmad et al., 2014a*; *Ahmad et al., 2014b*). Biochar as a carbon-rich, solid product of thermal decomposition of organic matter under a limited supply of oxygen and temperature <1,000 °C provide a wide range of positive properties for environmental and soil management. The behaviour of pyrogenic organic matter (charcoal, biochar, activated carbon) compared to non-pyrogenic organic matter included in soil is remarkably different. The main differences and changes in sorption characteristics were described by *Lehmann & Joseph (2015)*, not only found in sorption effectivity but also in different sorption mechanisms and in the reversibility of PAHs' sorption process. Sorption processes of PAHs by pyrogenic organic matter represent nonlinear processes with higher values of sorption distribution coefficients compared to non-pyrogenic organic materials. Interaction between biochar particles and organic xenobiotics can be described by various models. *Ahmad et al. (2014a)* and *Ahmad et al. (2014b)* suggested surface adsorption and electrostatic interactions as the main mechanisms of biochar contact with organic contaminants. The biochar high surface area and porous character play the main roles in these processes (*Yang et al., 2009*; *Lou et al., 2011*). *Chen, Zhou & Zhu (2008)* highlighted surface polarity and aromaticity as crucial parameters of organic pollutants removal from aqueous solutions by biochar-based sorbents. But in general, biochars can be considered as powerful sinks to retain PAHs (*Beesley et al., 2011*).

The influence of biochar on the sorption properties of soil for organic pollutants has been shown by several authors (*Spokas et al., 2009*; *Zhang et al., 2010*; *Chen & Yuan, 2011*). *Cornelissen & Gustafsson (2005)* showed that unburned coal carbon, black carbon, and amorphous organic carbon strongly influenced the sorption of phenanthrene in sediments. Additionally, *Hale, Luth & Crowley (2015)* considered the potentially positive but also the negative effects of biochar on the natural biodegradation of organic pollutants in terms of pollutant bioavailability, bioaccessibility, electron transfer and the availability of inorganic nutrients. Such intense sorption properties of biochar could modulate PAH degradation processes in soil by changing soil properties but also by decreasing availability of organic pollutants. Therefore, biochar can replace conventional remediation tools and become a low-cost alternative for activated carbon-based immobilisation amendments which combines several functions at the same time (*Beesley et al., 2011*). However, the detailed role of biochar in sorption and the immobilisation process of phenanthrene by biochar-amended soils is less known, and has to be experimentally clarified and tested in different soils. Particularly interesting for our work, (*Marchal et al., 2013*) and (*Sigmund et al., 2018*) have recently discussed the influence of sorption of phenanthrene in biochar on its degradation, resulting that sorption slows degradation. The former study used bacterial inoculation, so it needs validation in real soils, while the latter described that future research

should address soils and biochars with different properties to validate results and study interactions between components.

In relation to all the previous, the main aim of our work was to study the potential effect of biochars produced from three different feedstock on phenanthrene (Phe) sorption/desorption properties and on its mineralisation in two contrasting soils from the Western German region and pure sand. Our hypothesis is that biochars will be able to modify contaminant availability in soils, but the extent of the changes will depend on the soils and the biochars.

## MATERIALS AND METHODS

### Soils, sand and biochars

Soils were collected from two agricultural fields in the vicinity of Bochum, Germany. One was a soil (B) from a conventional agricultural field in Bottrop, 51°36′51.12″N, 6°54′12.43 W; the other was a soil (W) from an organic and conservation agriculture field in Witten, 51°27′10.24″N, 7°19′4.04″. The landowners, Bert Schulze-Poll, Witiko Ludewig and Hubertus Schulte-Kellinghaus, gave the permission to collect the samples from their agricultural fields. Commercial low-iron sand (Fisher Scientific Co., Hampton, NH, USA) of 40–100 $\mu$m size of grain was washed with HCl 10% and then with deionised water 10 times until the pH rose and stabilised (∼5.5). Three biochars from different origins were used in the experiment: (i) a pine woodchip biochar (PBC); (ii) an olive pruning biochar (OBC), and (iii) a rice husk biochar (RBC), all of them were produced by pyrolysis in a muffle furnace at 450 °C and a residence time of 15 min.

### Treatments

The biochars (size <1 mm) were incorporated in the tested soils and the sand at a rate of 1% (w/w); this is a realistic amendment as it corresponds approximately to a dose of 20 t ha$^{-1}$, considering 30 cm of soil depth and a soil density of around 1.6 t m$^{-3}$. After homogenizing the mixture, each sample was wetted to 60% of the water holding capacity and left to equilibrate for 15 days in polyethylene bags at room temperature (20–25 °C). All of the procedures and experiments (basic characterisation, batch interactions and soil incubations) were carried out with 3–4 replicates.

### Basic parameters in soils

Soil pH was analysed in a 1:5 soil:water extraction, shaken for 10 mins and left to sediment for 30 mins and measured in the supernatant with a bench pH-meter. DOC was analysed in a 1:10 soil:water extraction (during 4 h) and analysed with a Dimatoc 100 auto analyzer (Dimatec, Essen, Germany). Dehydrogenase activity was assessed according to *Von Mersi & Schinner (1991)* by weighing 1 g of dry soil and incubating it in a solution with 1 mL 0.4% 2-(p-iodophenyl)-3-(p-nitrophenyl)-5-phenyltetrazolium chloride (INT) and 0.05 mL 1% glucose for 24 h at 23 °C, the reaction was stopped by adding 10 mL methanol and the absorption in the supernatant analysed at 485 nm. A blank with no INT was used to correct background color in each sample. Respiration was assessed with a microtest: 0.3 g of soil was placed in a 96-deep well plate, closed overhead with another 96-well plate

which was filled by a $CO_2$ trap consisting of 1% agar and a pH indicator (Cresol red) that was analysed with a multiplate reader (TECAN INFINIT F200) after 6 h incubation. The changes in the indicator were related to soil respiration. FTIR spectra were analysed in a Bruker FTIR Spectrometer Tensor 27 HTS-XT. C, H and N were determined in a Vario Cube EL, Elementar Analysesysteme, Germany. For elemental analysis in biochar, elements were acid extracted according to Rajkovich et al. 2012 after dry ashing and measured with ICP-OES (Ciros CCD; Spectro Analytical Instruments, Kleve, Germany).

## Phenathrene sorption experiment

The experimental design has been adjusted according to OECD-Guideline 106 (2001): OECD Guideline for the testing of chemicals. Adsorption-Desorption using a batch equilibrium method. Organisation for Economic Co-operation and Development (OECD), Paris. 0.7 g equivalent dry soil was weighed in glass centrifuge tubes and 7 ml of Phenathrene (Phe) loaded solution at different concentrations: 1, 2, 5 and 10 μg Phe $g^{-1}$ soil. Each solution of Phe was prepared by dissolving the non-labelled commercial solid product (99 % purity; Sigma-Aldrich Chemie GmbH, Steinheim, Germany) in ethanol, while the $^{14}$C-Phe was already in solution (99.7% purity, Campo Scientific GmbH, Germany). The solutions were prepared by keeping the proportion of ethanol <0.1%, adding the specific amount of non-labelled Phe and always ∼1,000–1,600 Bq $^{14}$C Phe $g^{-1}$ soil, in a background solution of 10 mM of $CaCl_2$ and 1.5 mM $NaN_3$. Tubes were shaken in an overhead shaker at 100 rpm for 24 h and afterwards they were centrifuged at 2700g for 15 min and 1 ml aliquot of the supernatant was sampled, processed and analysed in a beta counter by liquid scintillation counting (Tri-Carb 2800TR analyzer; PerkinElmer, Waltham, MA, USA). The interaction time was previously defined in a preliminary batch interaction using 10 μg Phe $g^{-1}$ soil as described before, but taking an aliquot of the supernatant at 7, 24 and 48 h of interaction. The radioactivity was also analysed in the beta counter.

## Phenanthrene desorption experiment

For this experiment we selected the samples loaded with Phe from the sorption experiment after the interaction. We used the method reported in *Shchegolikhina & Marschner (2013)*, with the only modification of replacing up to 90% of the solution between each step. So we extracted the Phe retained in the soil during the sorption experiment for the dose 1 μg Phe $g^{-1}$ soil with six sequential cycles of water-ethanol. The first three extractions were performed with deionised water, with durations of 3, 3 and 7 h. The next three were done with ethanol absolute, shaking each of them for a duration of 1 h. After each cycle, the tube was centrifuged at 2700 g for 15 min and 1 ml aliquot mixed with scintillation cocktail and analysed in the beta counter. The remaining supernatant was replaced in its majority by new solution.

## Phenanthrene mineralisation experiment

For this experiment, we used the experimental setup described in *Shchegolikhina & Marschner (2013)*. 20 g of soil were placed inside a closed vessel. This soil was spiked with 1% Phe-loaded sand to obtain a final dose in the soil of 1 μg Phe $g^{-1}$ and 40 Bq $^{14}$C Phe $g^{-1}$. Each closed vessel also contained a 1 M KOH trapping solution that was renewed

twice during the incubation. Vessels were placed inside a warm bath (30 °C) and incubated for four weeks. A 1 mL aliquot was sampled from each trap at one, two and four weeks and mixed with scintillation cocktail for radioactivity assessment in a beta counter. At the end of the incubation, soil samples were again subjected to a desorption as described previously.

## Data processing and statistical analyses

We calculated in the kinetics experiment two parameters, $K_d$ and $K_{oc}$.

The distribution coefficient ($K_d$) was calculated using the equation:

$$K_d = \frac{V \cdot (c_0 - c_{eq})}{w \cdot c_{eq}}$$

and the carbon–normalised distribution coefficient ($K_{oc}$) was calculated using the equation:

$$K_{oc} = \frac{V \cdot (c_0 - c_{eq})}{w \cdot f_{oc} \cdot c_{eq}}$$

where $c_0$ is the initial concentration of the compound ($\mu$g mL$^{-1}$), $c_{eq}$ is the aqueous phase concentration of the xenobiotic at the end of the sorption–desorption experiment ($\mu$g mL$^{-1}$), $V$ is the volume of the aqueous phase (mL), $w$ is the mass of soil (g), and $f_{oc}$ is the weight fraction of organic C in the soil.

For the sorption experiment, the data was evaluated by the Freundlich sorption model. In the nonlinear form, the Freundlich isotherm behaves according to the next equation:

$$Q_{eq} = KC_{eq}^{(1/n)}$$

where $Q_{eq}$ is the amount of sorbed Phe at equilibrium (mg g$^{-1}$), $K_F$, $n$ are the Freundlich empirical constants characterizing parameters and intensity of sorption process ($K_F$ as L g$^{-1}$), $C_{eq}$ is the Phe equilibrium concentration in solution (mg L$^{-1}$).

Linear form of Freundlich isotherm is described as:

$$\log Q_{eq} = \log K_F + \frac{1}{n}\log C_{eq}$$

For the desorption experiment, the percentage of Phe extracted in each cycle was estimated as:

$$E_n(\%) = 100 \cdot \frac{s_n}{s_0}$$

where $s_n$ is the total radioactivity (Bq) in the extracting solution at the end of one extraction cycle. $s_0$ is the total radioactivity in the tube: in the sorption experiment the radioactivity is bound to the soil and contained in the 0.5 mL of solution; in the incubation experiment, the radioactivity in the soil.

For each aliquot of the degradation experiment when the respired $CO_2$ was collected in the KOH solution the percentage of mineralised xenobiotic $M_n$ was calculated from the relationship:

$$M_n(\%) = 100 \cdot \frac{c_n}{c_0}$$

where $c_n$ (Bq) is the total radioactivity detected in the KOH solution at 6, 14 or 28 days and $c_0$ (Bq) is the initial radioactivity of the soil sample spiked with Phe.

The data was statistically processed with SPSS® 16.0. Data was checked for normality and homocedasticity to decide the appropriate procedure. When variances were homogenous, we tested the influence of biochar with a one-way ANOVA and a Tukey's test to separate groups. With heterocedastic values, we used Games-Howell as post-hoc test. Linear regression analysis between two variables to estimate sorption isotherm parameters was done with Excel. A principal component analysis (PCA) was conducted to elucidate the correlation of single soil mineralisation and sorption parameters. The data which were not showing normal distribution were log-transformed before conducting PCA. Further the data were rotated by Varimax rotation. pH was excluded from the procedure due to reducing Kaiser-Meyer Olkin criteria.

## RESULTS AND DISCUSSION

### Physico-chemical properties of soil, amendments and pH influence

The basic characteristics of the soils differed at the initial time of incubation. The initial pH of the Witten soil sample (6.6) was slightly more alkaline compared to the Bottrop sample (6.3); the sand pH was around 5.4. Physicochemical characterisation revealed soil organic C contents of 2.2% C in W and 2.6% in B, while the sand had very low content (<0.1%); C/N values were 12.5 for B and 15.0 for W soil. Textural analysis showed characteristic values of sand, silt and clay soil components respectively as 6.7%, 85.6% and 7.6% for Witten soil (silt) and 38.3%, 54.0% and 7.7% for Bottrop soil (silt loam).

The main physico-chemical characteristics of the applied biochar amendments are shown in Table 1. A comparison of the applied materials shows a general difference in pH of the biochar-based amendments. Therefore, the liming potential decreases in order OBC>PBC>RBC. Total carbon content determination was confirmed as >80% C in OBC and PBC. On the other hand, RBC material showed C content as <50% and therefore would not be classified as a biochar following the *EBC (2012)* but as pyrogenic carbonaceous material. Remarkably, this RBC showed a high content of ashes (∼50%). OBC had higher ash content that PBC, mainly alkaline cations, which may explain the higher pH in the former.

Biochar surface functionality can be demonstrated by FTIR spectra evaluation (*De la Rosa et al., 2014*). The characteristics absorption peaks were observed in spectra of OBC, PBC and RBC at wavelengths of 3,040 $cm^{-1}$ attributed to the vibration of C-H groups in aromatic structures and wavelengths in the range 2,850–2,950 $cm^{-1}$ attributed to the C-H stretching vibration of aliphatic CH, $CH_2$ and $CH_3$ groups. Strong peaks in the vicinity of wavelengths 1,700 $cm^{-1}$, 1,600 $cm^{-1}$ and 1,450 $cm^{-1}$ were attributed to asymmetric and symmetric stretching vibration of C=O, C=C and C-H in aromatic and aliphatic structures. The absorption peaks at $\sigma$ 1,050–1,100 $cm^{-1}$ represent the C-O and C-H bending in biochar composition. Additionally C-H stretching vibration in aromatics at wavelengths 800–900 was reflected. Similar peaks were found and described in detail in studies *De la Rosa et al. (2014)*, *Frišták et al. (2015)* and *Tang et al. (2015)*. Comparing the sample's spectrum (Fig. 1), OBC and PBC seem to have more aliphatic character, which is evident in the more intense signal observed at 2,850–2,950 $cm^{-1}$ and 1,450 than in

**Table 1  Basic physico-chemical characteristics of olive pruning-derived biochar (OBC), pine woodchips-derived biochar (PBC) and rice husk-derived biochar (RBC).**

|  | OBC | PBC | RBC |
|---|---|---|---|
| pH | 9.3 | 7.5 | 6.5 |
| Ash content (%) | 10.0 | 1.8 | 52 |
| C (%) | 82.9 | 85.2 | 43.1 |
| N (%) | 1.21 | 0.37 | 0.6 |
| H (%) | 2.67 | 2.78 | 2.4 |
| Mg ($mg\,kg^{-1}$) | 2,088.3 | 574.0 | 968.0 |
| Fe ($mg\,kg^{-1}$) | 496.3 | 199.7 | 1,803.0 |
| Na ($mg\,kg^{-1}$) | 725.3 | 335.3 | 264.0 |
| K ($mg\,kg^{-1}$) | 9,158.5 | 1,708.2 | 2,661.0 |
| P ($mg\,kg^{-1}$) | 1,463.8 | 147.7 | 546.0 |

RBC, with OBC being the most aliphatic (the most intense signal at $1,450\ cm^{-1}$). OBC and PBC seem also to have more predominance in polysaccharides, with a more intense signal at $1,050–1,100\ cm^{-1}$ than RBC. Contrastingly, RBC had a more intense signal at $800–900\ cm^{-1}$, which may indicate higher aromaticity.

The determination of pH values of amended Witten and Bottrop soils with 1% (w/w) PBC, OBC, RBC after short-term incubation showed a rapid increase in the case of all amendments and both soil samples. OBC and RBC addition increased the pH values of the Witten and Bottrop soils significantly by the end of the incubation ($P < 0.05$). In comparison with the control sample, 1% OBC amendment increased the pH of the W and B soil samples, respectively, by 0.5 and 0.3 units. Amendments of 1% RBC caused a pH increase of 0.2 units for the Witten and 0.5 units for the Bottrop soil. The PBC amendment has not confirmed statistical significance in the altering of both soil reactions. The control samples with sand also showed the highest influence of OBC and RBC amendment on change of pH value after short-term incubation (Fig. 2). The more intense alkalinisation in RBC and OBC treatments could be related to the higher alkaline cation content in these biochars (Table 1).

## The influence of carbonaceous amendments on biochemical soil properties

Dissolved organic carbon (DOC) represents one of the most important factors of soil sorption complex as it may act as a mobile sorbent and thus reduce the apparent sorption to the solid phase (*Marchal et al., 2013*). The amount of water extractable DOC in the soil samples and sand samples was increased by biochar amendments (Fig. 3). Control sand samples showed an increasing DOC in the order OBC<RBC<PBC. The content of DOC in the amended Witten soil samples showed a similar trend. Here, the highest concentration of dissolved organic carbon ($>60\ mg\,kg^{-1}$) was found in the PBC-amended samples. On the other hand, the Bottrop soil samples showed a different trend: the PBC-amended soil showed lower DOC than the RBC-amended soil, which reached $>65\ mg\ DOC\ kg^{-1}$.

Soil enzymatic activity represents a crucial parameter and sensor in the assessment of soil amendment influences on physico-chemical properties and soil sorption potential

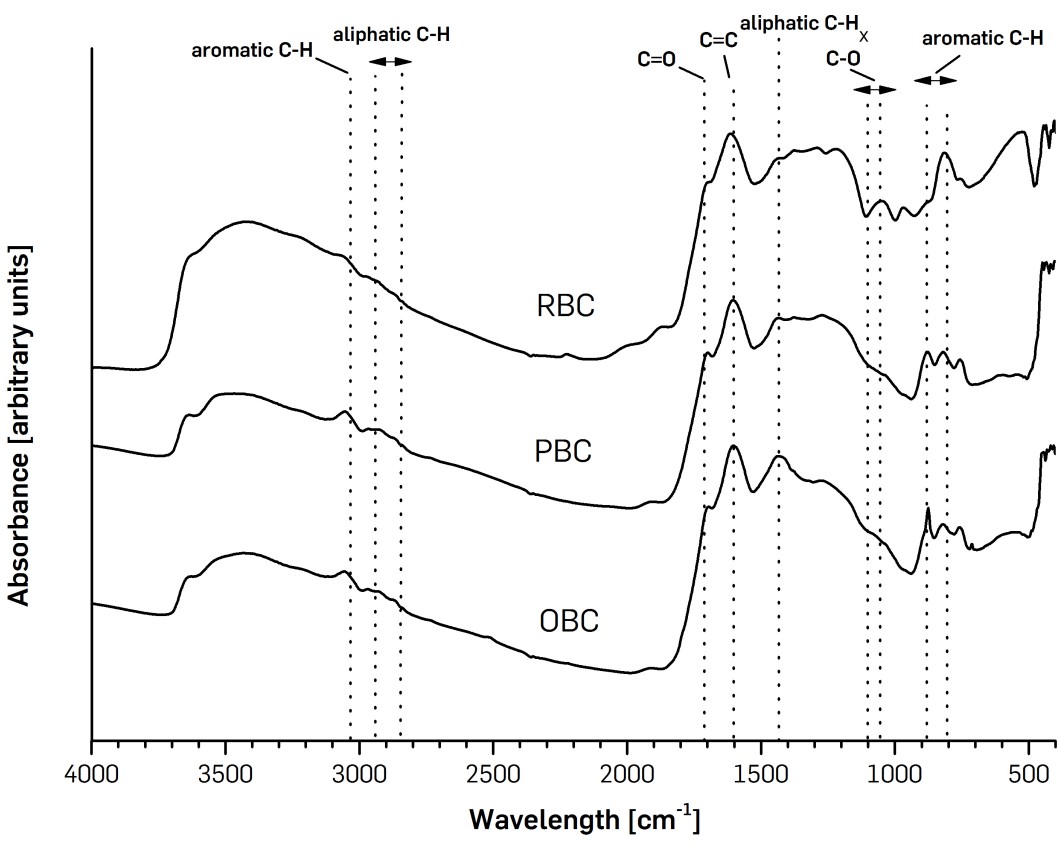

**Figure 1** **FTIR spectra of pinewoodchips-derived biochar (PBC), olive pruning-derived biochar (OBC) and ricebiochar (RBC) samples.** Each curve represents one biochar. Range of representative peaks are indicated.

(*Tica, Udovic & Lestan, 2011*). The effect of biochar amendments on soil enzyme activities varies in terms of different properties of biochar (feedstock, pH, micro- and macro-nutrients content) and different soil characteristics (*Watzinger et al., 2014*). Additionally, the altering of microbial activity can be caused by biochar effect on soil microbial community or enzymatic processes (*Lee et al., 2009*). *Garau et al. (2007)* described the main role of dehydrogenases as an indicator of total microbial activities in response to soil amendment incorporation. The determination of dehydrogenase activity (DHA) in soil samples studied in our work revealed higher values for unamended soil Bottrop compared to Witten (Fig. 4). The biochar incorporation into Bottrop soil samples showed no effect in the case of PBC and OBC, while RBC decreased DHA ($P < 0.05$). In the case of the Witten soil samples, PBC and OBC enhanced DHA ($P < 0.05$), while RBC did not change this parameter in comparison to the sample without any addition. As expected, the DHA in sand was nearly not detectable.

Basal and substrate-induced soil respiration is widely used to estimate the effect of amendment on soil organisms (*Tica, Udovic & Lestan, 2011*). Our results showed inhibition of soil respiration in the case of RBC-amended Witten soil samples ($P < 0.05$)

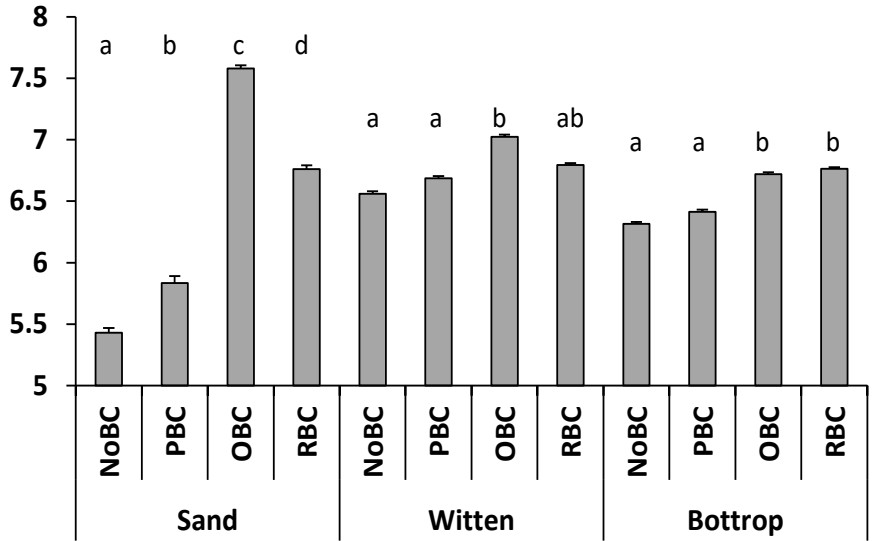

**Figure 2 pH values of sand and soil samples with different biochar treatment.** Non-amended (NoBC) samples (sand, Witten soil, Bottrop soils) and samples amended by 1% (w/w) pine woodchips-derived biochar (PBC), olive pruning-derived biochar (OBC) and rice biochar (RBC). The results are presented as means $\pm$ SD ($n = 3$). The different letters denote significant differences ($p < 0.05$) between means.

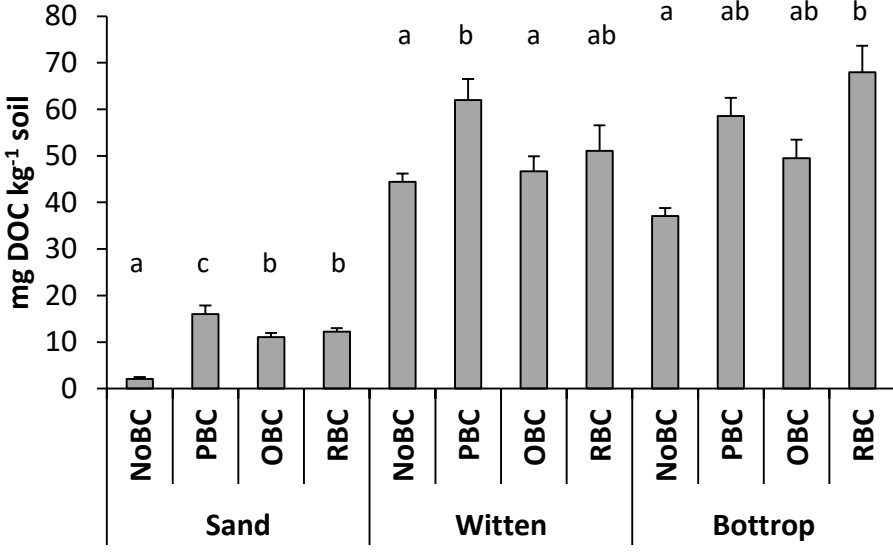

**Figure 3 Dissolved organic carbon in sand and soil samples with different biochar treatments.** Non-amended (NoBC) samples (sand, Witten soil, Bottrop soil) and samples amended by 1% (w/w) pine woodchips-derived biochar (PBC), olive pruning-derived biochar (OBC) and rice biochar (RBC). The results are presented as means $\pm$ SD ($n = 3$). The different letters denote significant differences between means.

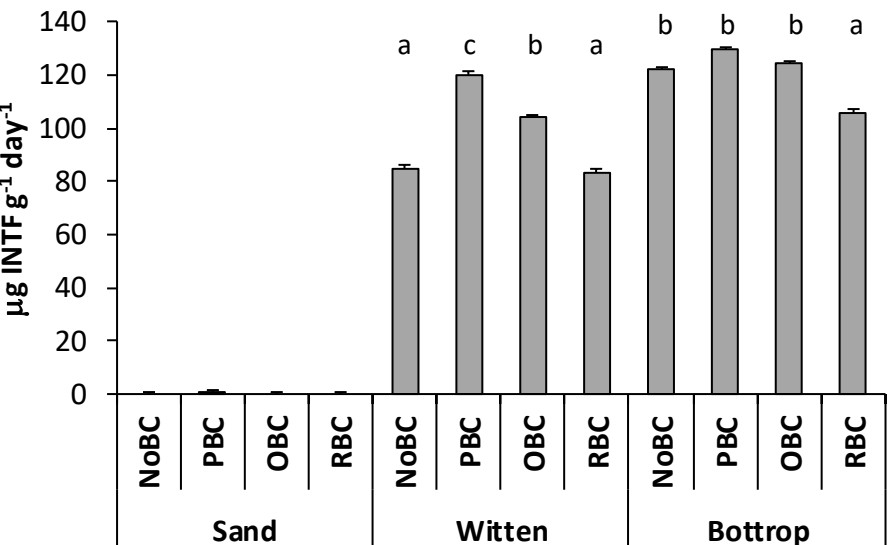

**Figure 4** **Dehydrogenase activity (m g INTF g$^{-1}$ day$^{-1}$) in sand and soils with different biochar amendments.** Non-amended (NoBC) samples (sand, Witten soil, Bottrop soil) and samples amended by 1% (w/w) pine woodchips-derived biochar (PBC), olive pruning-derived biochar (OBC) and rice biochar (RBC). The results are presented as means ± SD ($n = 3$). The different letters denote significant differences ($p < 0.05$) between means.

and no statistically significant effect of PBC and OBC on the respiration activity in the same soil (Fig. 5). On the other hand, OBC-amended Bottrop soil had 50% higher respiration compared to non-amended Bottrop soil ($P < 0.05$). The effects of PBC and RBC amendments on Bottrop soil respiration appeared statistically insignificant. Over all, the Witten soil showed the highest mean respiration with 1.39 μg $CO_2$-C g$^{-1}$ h$^{-1}$ followed by Bottrop soil (0.61 mg $CO_2$-C g$^{-1}$ h$^{-1}$) and the sand control soil (0.14 μg $CO_2$-C g$^{-1}$ h$^{-1}$).

## Sorption process of $^{14}$C-labelled phenathrene

Obtained kinetic data illustrates the effect of contact time on the sorption process of Phe (Table S1) by PBC, OBC and RBC amended soils. Sorption parameters of $K_d$ and $K_{oc}$ increased significantly ($P < 0.01$ in all cases, $P < 0.1$ for $K_{oc}$ in Bottrop) in all soils from seven to 24 has obtained by non-parametric tests. After 24 h the sorption system reached equilibrium, $K_d$ and $K_{oc}$ did not differ from those analysed at 48 h. Similar equilibrium time was reached by *Zhang et al. (2010)* for sandy and silty loam soils amended by *Pinus radiate*-derived biochar and phenathrene sorption. Our results showed increased affinity of Phe to the biochar-amended Witten soil in the order of RBC>OBC>PBC>NoBC and of OBC>RBC>PBC>NoBC for the pure sand. In contrast, $K_d$ values for Bottrop soil samples decreased with biochar additions in the order NoBC>OBC>PBC>RBC. As expected, sorption in pure sand was very weak, but with biochar application increased in all cases, following these order: OBC>RBC>PBC>NoBc. For the following experiments an equilibration time 24 h was used.

The sorption experiments with initial Phe concentrations ranging from 1–10 μg Phe g$^{-1}$ soil show a clear, statistically significant positive effect of all carbonaceous amendments on
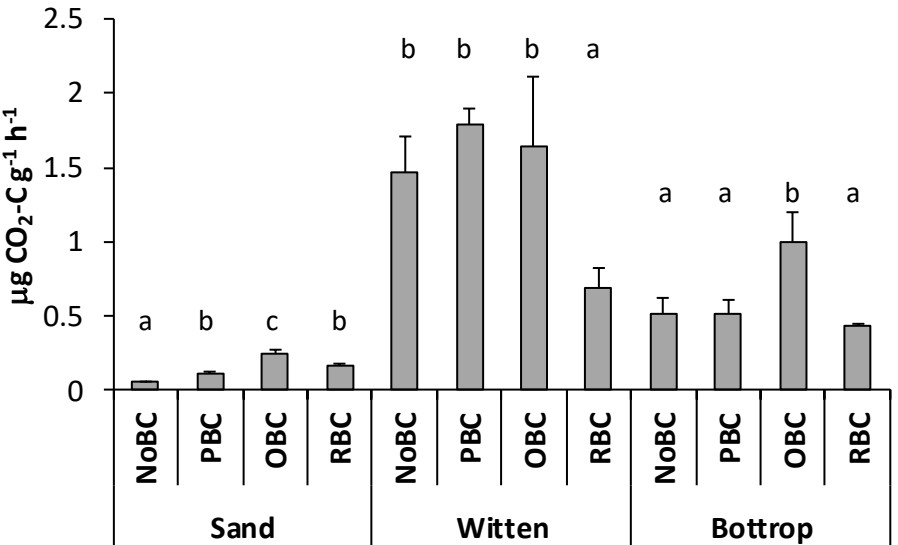

**Figure 5** **Basal respiration in sand and soil samples with different biochar amendments.** N on-amended (NoBC) samples (sand, Witten soil, Bottrop soil) and samples amended by 1% (w/w) pine woodchips-derived biochar (PBC), olive pruning-derived biochar (OBC) and rice biochar (RBC). The results are presented as means $\pm$ SD ($n = 4$). different letters denote significant differences ($p < 0.05$) between means.

$K_d$ and $K_{oc}$ in the Witten soil samples while in the Bottrop soil, no significant treatment effects on Phe sorption were detectable.

The fitting of the obtained Phe sorption data with the Freundlich adsorption model (Table 2) were adequate in the case of both non-amended and biochar-amended soil samples ($R^2 > 0.65$). *Zhang et al. (2010)* also showed good applicability of the Freundlich adsorption model to describe Phe sorption data by sandy and silty soil samples amended by *Pinus radiata*—derived biochar. Parameters obtained from the analysis revealed the main effect of PBC, OBC and RBC on sorption efficiency of sand, Witten and Bottrop soil samples for Phe, being $K_F$ an approximation of the soil to solution partition. Higher $K_F$ values mean higher sorption in the solid phase, i.e., immobilisation (*Igwe & Abia, 2007*). Also $K_F$ was generally in good agreement with $K_d$ in the kinetic experiment (Table S1) and the $K_d$ calculated for this experiment (Fig. S1). PBC showed the lowest positive effect to improve the sorption potential of pure sand, Witten and Bottrop soils, while OBC and RBC increased the sorption potential of all three.

The value of $1/n$ parameter represents a joint measure of both the relative magnitude and diversity of energies associated with a particular sorption process (*Weber, McGinley & Katz, 1992*). $1/n = 1$ shows linear sorption process and therefore equal sorption energies for all sorption sites. Generally, linear sorption process is characteristic for low concentrations of sorbate and low amounts of sorbent as well. Value $1/n > 1$ represents a concave isotherm, where the marginal sorption energy increases with increasing surface area (*Delle, 2001*). The process can be characterized by intesive sorption of sorbate, relatively strong intermolecular attraction within layers of sorption material and mainly non-functional nature of sorbent.

**Table 2  Freundlich isotherm parameters of Phe sorption ($1/n$, $K_F$ and $R^2$) in sand and soils with different biochar treatments.** N on-amended (NoBC) samples (sand, Witten soil, Bottrop soil) and samples amended by 1% (w/w) pine woodchips-derived biochar (PBC), olive pruning-derived biochar (OBC) and rice biochar (RBC).

| | Amendment | $1/n$ | $K_F$ | $R^2$ |
|---|---|---|---|---|
| Sand | NoBC | – | – | 0.12 |
| | PBC | $1.06 \pm 0.37$ | $0.60 \pm 0.83$ | 0.48 |
| | OBC | $0.76 \pm 0.16$ | $2.12 \pm 0.25$ | 0.70 |
| | RBC | $0.31 \pm 0.08$ | $2.72 \pm 0.14$ | 0.65 |
| Witten | NoBC | $0.72 \pm 0.13$ | $2.42 \pm 0.17$ | 0.76 |
| | PBC | $0.71 \pm 0.12$ | $2.51 \pm 0.15$ | 0.78 |
| | OBC | $0.99 \pm 0.21$ | $2.50 \pm 0.24$ | 0.68 |
| | RBC | $0.63 \pm 0.08$ | $2.63 \pm 0.10$ | 0.86 |
| Bottrop | NoBC | $0.71 \pm 0.17$ | $2.63 \pm 0.18$ | 0.65 |
| | PBC | $0.87 \pm 0.08$ | $2.48 \pm 0.09$ | 0.72 |
| | OBC | $0.65 \pm 0.08$ | $2.79 \pm 0.08$ | 0.86 |
| | RBC | $0.56 \pm 0.06$ | $2.80 \pm 0.06$ | 0.90 |

On the other hand, $1/n < 1$ represents a convex isotherm, where the marginal sorption energy decreases with increasing specific and nonspecific surface area (*Delle, 2001*). It occurs when the competition of sorbate for active sorption sites is negligible or the sorbate represents a planar molecule. In this case, the mobility of a selected sorbate in soil matrix can be significantly greater for the higher concentrations. Our study showedthe parameter $1/n$ varied among all the soils and treatments and this is indicative of the sorption, with $1/n \sim 1$ having a linear sorption in all the range of Phe in solution and $1/n < 1$ showing concave decreasing sorption as Phe increases in the solution. For instance, sand and soils with PBC showed in general values closer 1, indicating a C-type isotherm, where the physisorption should predominate and the relative affinity of the Phe for the solid remains constant in the tested range. In contrast, pure sand and soils with RBC showed the lowest values of this parameter ($1/n < 0.61$), which indicates that sorption efficiency decreased at the highest range tested. In between, OBC treatments showed $1/n = 1$ in Witten soil and in Bottrop soil 0.65. As a general rule, therefore, the two stronger sorbents (OBC and RBC) in the soils showed a decreasing curvature of sorption as Phe increases, indicating that they are highly efficient at lower doses of Phe. This decreasing curvature was especially obvious in RBC, the material with the highest Fe concentration (Table 1), and Fe has been reported as responsible for main changes in sorption properties of several sorption materials (*Gao et al., 2006*). Also, various DOC values can affect sorption properties (*Pan et al., 2007*) of amended soil samples and thus change the sorption potential for Phe. The sorption non-linearity of Freundlich isotherm increased in order OBC<PBC<RBC for Witten and PBC<OBC<RBC for Bottrop. RBC amendment showed most significant effect on sorption non-linearity what can be discussed the origin of used material. Content of total C and ash content (Table 1) revealed the ability of potentially high non-carbonized fraction which can significantly influence the heterogeneity of amended soil samples and subsequently sorption process as well. Additionally the various distribution of active adsorption centres can be

discussed. *Sigmund et al. (2017)* showed and detail characterized the role of non-carbonized fractions of biochar in sorption processes of organic molecules. Authors emphasized the correlation between increased non-linearity of Freundlich isotherm, more heterogeneous distribution of available sorption sited energies and hence shift towards adsorption process.

To evaluate the additive effect of biochars on the sorption of Phe in soils, we compared the experimental average sorption ($K_d$ and $K_{oc}$) to the calculated sorption if each biochar' effects were completely additive (Fig. S2). The slope was very close to 1, so in general the effects are additive, but there are two samples that are consistently above the line, which belongs to Bottrop with OBC and RBC. This means those two biochars were not as efficient as expected in sorbing Phe in the Bottrop soil, which was in this case concomitant to an increase in soil pH in this soil with both biochars. Changes in the pH can modify some characteristics of the soil system as a sorbent (functionality, charge of soil mineral components, changes in humic substances fate, ionic strenght etc.). In this study, soil samples represent heterogenous matrixes with a wide range of components, so we would need further and more extensive studies to disentangle the role of pH in Phe sorption in these soils.

## Desorption of [14]C-labelled phenantrene

For this experiment, soils previously loaded with 1 µg Phe g$^{-1}$ from the sorption experiment were subjected to a sequential desorption. For all cases, the recovery of the Phe from the samples ranged from 90–99%, being a good representative extraction method (*Shchegolikhina & Marschner, 2013*). As expected, sand sorption of phenantrene was very labile, so almost the total Phe was extractable with water (Fig. 6). Sand with PBC also showed an intense desorption (50%) in water, indicating low affinity of Phe for PBC biochar. On the other hand, Phe was more intensively sorbed by sand with OBC and RBC (desorption ∼15%), reinforcing that these biochars act as good and stable sinks for Phe. In both soils, Witten and Bottrop, only a small portion of Phe was water extractable (4–10%), independently of biochar application. Sorption of Phe in soils is usually quite intense, mainly mediated by the soil organic matter, where Phe can be efficiently retained (*Kleineidam et al., 1999*). It is noteworthy that similar to the pure sand, OBC and RBC also decreased the water extractable proportion of Phe in both soils, which is inversely related to the higher sorption potential ($K_d$ and $K_{oc}$) in these treatments (Fig. S1).

## Mineralisation of [14]C-labelled phenantrene

For this experiment, soils spiked with 1 µg Phe g$^{-1}$ soil were incubated for 29 days (Fig. 7). After 29 days, the cumulative Phe mineralisation in the sand samples was almost negligible while it reached 12–15% in the soils (Fig. 7). In the sand, RBC slightly increased Phe mineralisation from 0.3% in the control to 1% ($P < 0.05$). In the Witten soil, OBC and RBC both decreased Phe mineralisation ($P < 0.05$), while PBC had no effect. In the Bottrop soil, no significant effects of the biochar amendments on Phe mineralisation were found although RBC and OBC also showed the lowest Phe mineralisation. *Marchal et al. (2013)* found that mineralisation rate was suppressed in soil where Phe was not easily desorbed. In agreement with this, the reduction in desorption of Phe when adding RBC and OBC to

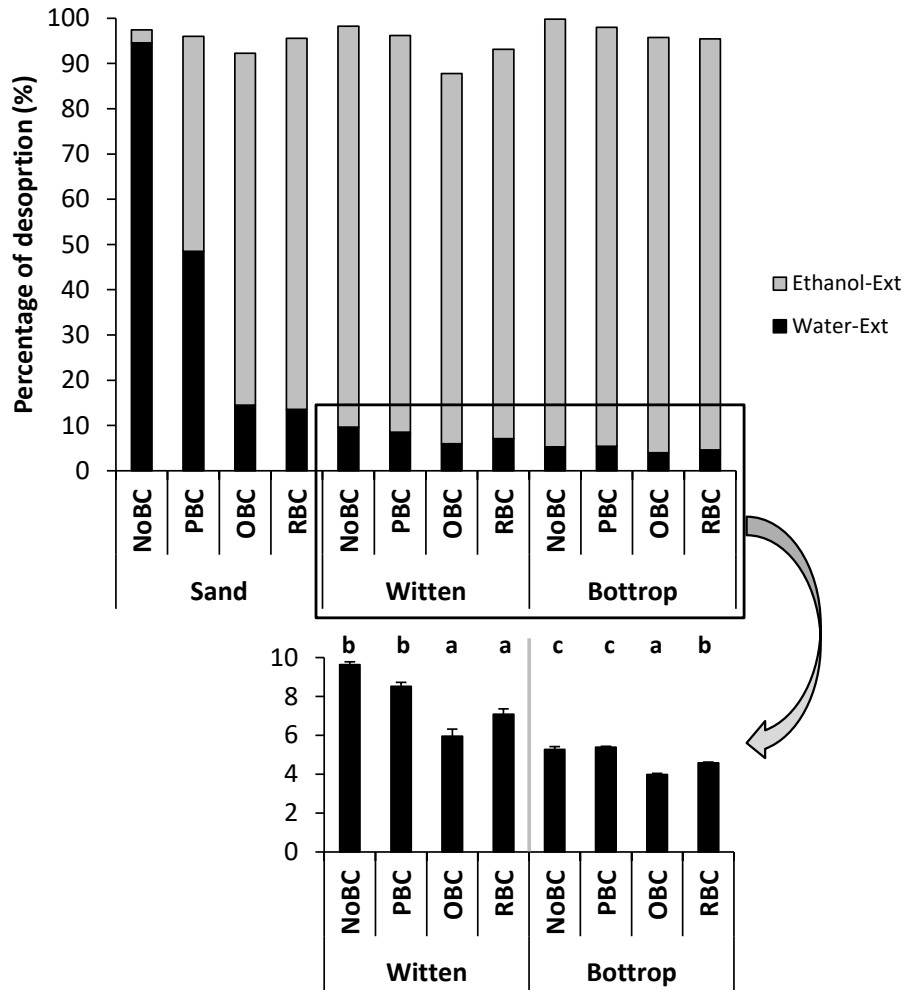

**Figure 6** **Desorption of Phe (as % of Phe in each sample) from sand and soils with different biochar treatments.** Non-amended (NoBC) samples (sand, Witten soil, Bottrop soil) and samples amended by 1% (w/w) pine woodchips-derived biochar (PBC), olive pruning-derived biochar (OBC) and rice biochar (RBC). Samples from sorption experiment (1 mg Phe g$^{-1}$ soil replicates) were used. For each extraction, with water and with ethanol, three subsequent fractions were produced and combined into one for each extractant. The results are presented as means $\pm$ SE ($n = 3$). The different letters denote significant differences ($p < 0.05$) between means.

Witten soil occurred at the same time as a reduction in Phe mineralisation. In Bottrop we did not find any impact of biochars in mineralisation, but also the reduction of desorption caused by any biochar was lower than in Witten. Contrastingly, in the sand system, biochars enhanced Phe mineralisation, most likely by providing labile compounds present in fresh biochars (*Wang, Xiong & Kuzyakov, 2016*), as reflected in the increased DOC (Fig. 3) which stimulated microbial activity (Fig. 5). Interestingly, OBC increased microbial respiration by a factor of 5 but failed to increase Phe mineralization, while RBC only increased microbial respiration twofold relative to the control but increased Phe mineralization threefold. So, there is no direct relationship between Phe mineralisation and microbial activity in the

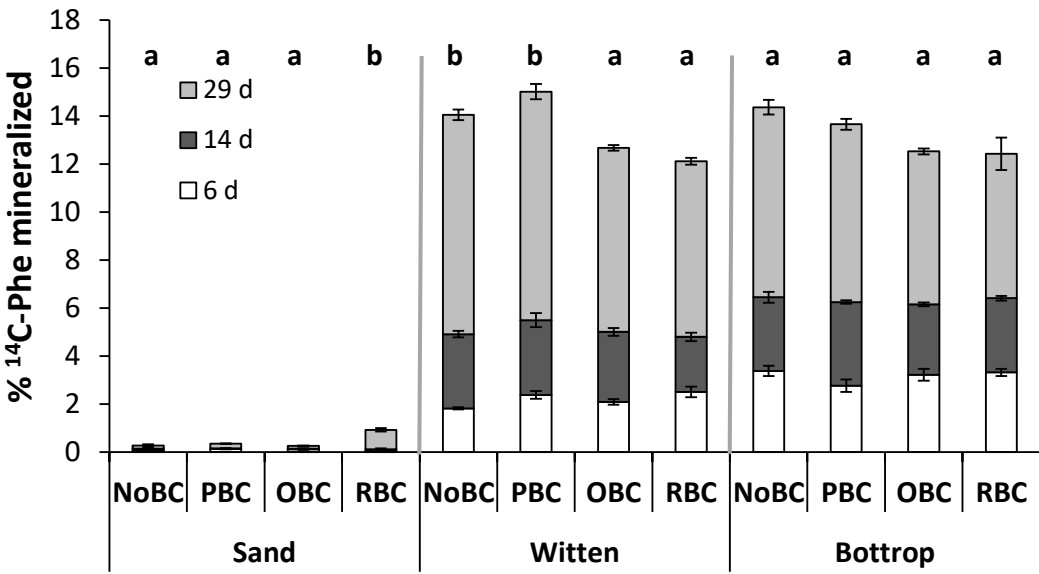

**Figure 7** **Mineralisation of Phe (as % of the initial Phe in each sample) in sand and soils with different biochar treatment.** Non-amended (NoBC) samples (S, sand; W, Witten soil; B, Bottrop soil) and samples amended by 1% (w/w) pine woodchips-derived biochar (PBC), olive pruning-derived biochar (OBC) and rice biochar (RBC). Soils were spiked with Phe at a dose 1 m g Phe g soil and incubated during 29 days. The results are presented as means ± SE ($n = 3$). The different letters denote significant differences ($p < 0.05$) between means.

biochar-amended sand samples. But the comparison between the sand and soil samples clearly shows, that Phe mineralisation is much higher in with the biologically more active soil samples.

Regarding the temporal trend, Fig. 7 shows that Phe minaeralisation in Bottrop soil was faster than in Witten during the first six days of incubation, which may indicate different availability of Phe for organisms between soils or different initial microbial Phe-degrading population abundance. From six days on, Witten and Bottrop showed similar rates of mineralisation.

## Integrative comments
The principal component analysis elucidates the correlation between different soil parameters (Table 3). The PCA generates two different factors which explain 89% of the total data variance. Interestingly, the $K_d$ value is included in both factors with a similar factor loading. In the first factor, Phe mineralisation is strongly connected to microbial activity with the parameters basal respiration and dehydrogenase activity which shows that higher microbial activity favours Phe mineralisation. Since DOC is also shows a high loading in this factor, this indicates that it may act as an important substrate for microbial activity (*Marschner & Kalbitz, 2003*) and may also increase Phe availability. The separation of phenantrene mineralisation and the sorption measures ($K_{oc}$ and water extractable $^{14}C$ (−0.829)) shows that Phe-mineralisation might be impacted stronger by microbial properties than sorption/desorption potential in the analysed treatments. As expected,

**Table 3** Factor loadings, explained variance and Kaiser-Meyer-Olkin (KMO) criteria of the principal component analysis (PCA) with Varimax rotation for soil properties over all treatments.

| Parameter[a] | Factor 1 | Factor 2 |
|---|---|---|
| Respiration ($\mu$g $CO_2$-C $g^{-1}$ $h^{-1}$) | **0.936** | 0.286 |
| DOC (mg $kg^{-1}$) | **0.847** | 0.136 |
| Dehydrogenase (g INF $g^{-1}$ $d^{-1}$) | **0.837** | 0.402 |
| $^{14}$C-Mineralisation (%) | **0.799** | 0.312 |
| Koc-values | 0.123 | **0.986** |
| Water extractable $^{14}$C (%) | −0.526 | **−0.829** |
| Kd-values | 0.693 | **0.709** |
| **Explained variance (%)** | **74.5** | **14.5** |
| KMO-criteria | 0.68 | |

**Notes.**

[a]Remarks: $^{14}$C-mineralisation, water extractable $^{14}$C, Kd-values and respiration data were log-transformed to reach normal distribution of the dataset.

sorption parameters ($K_d$ and $K_{oc}$) are inversely related to water extractability, with the higher sorption potential, the lower water extractability of Phe. This indicates that materials with high sorption potential are less likely to transferring the Phe to waters, stabilising the Phe in the solid phase.

As it was expected from the usual affinity of biochars to retain Phe, 1% biochar efficiently enhanced Phe sorption in pure sand. However, in both tested soils, which had a relatively high organic matter content (>2% SOC), the biochars were not able to remarkably enhance sorption in soil, likely due to the low application dose (see Fig. S2). From this findings, we can expect that in soils with low organic matter (below 1% of SOC), this biochar application dose will be efficient when Phe immobilisation is needed. In soils with higher SOC, like Witten or Bottrop, 1% biochar will not further immobilise Phe, but it will decrease the soluble fraction of Phe (Fig. 6), which is also positive to Phe immobilisation in soil. A total of 1% application of biochar would correspond to 20 t per ha, which is a realistic dose for the field. For soils with >1–2% SOC, doses of 2–3% of biochar should be tested as they probably will increase sorption of Phe but being still realistic doses to be applied in the field (20–60 t $ha^{-1}$). Higher doses are probably difficult to implement at a real scale (*Piccolo, Pietramellara & Mbagwu, 1996*).

## CONCLUDING REMARKS

This study highlights the soil- and biochar-specificities affecting Phe sorption in soils, showing how contrasting biochars are able to differently sorb this organic compound and affect mineralisation. For the immobilisation of Phe in soils, rice- (RBC) and olive pruning-derived biochar (OBC) were most appropriate to enhance Phe sorption and prevention of leaching, probably also decreasing its bioavailability. In terms of mineralisation, the Phe mineralisation was always <15% during one month at ideal conditions, and RBC and OBC decreased the mineralisation of Phe, and this mineralisation was basically microbiologically-driven. Thus, sorption potential is inversely related to water-desorption and mineralisation, and the appropriate biochar should be selected depending of the

goal (i.e., immobilisation vs. degradation). As each biochar behaved slightly differently between soils, soil-biochar interactions should be taken into account in future research and caution is needed when extrapolating to other soils or to field conditions, because the experiment used manipulative conditions (i.e., not studying the changes in the soil biota under manipulative conditions in respect to the field).

### Funding
This work was partly supported by Spanish MINECO, project number CTM2013-48697-C2-2-R. COST Action TD1107 funded Eduardo Moreno Jiménez during his experiments in Germany. There was no additional external funding received for this study. The funders had no role in study design, data collection and analysis, decision to publish, or preparation of the manuscript.

### Grant Disclosures
The following grant information was disclosed by the authors:
Spanish MINECO: CTM2013-48697-C2-2-R.
COST Action: TD1107.

### Competing Interests
The authors declare there are no competing interests.

### Author Contributions
- Eduardo Moreno Jiménez conceived and designed the experiments, performed the experiments, analyzed the data, prepared figures and/or tables, authored or reviewed drafts of the paper, approved the final draft.
- Sara Aceña-Heras, Vladimir Fristak and Stefanie Heinze analyzed the data, prepared figures and/or tables, authored or reviewed drafts of the paper, approved the final draft.
- Bernd Marschner conceived and designed the experiments, analyzed the data, contributed reagents/materials/analysis tools, authored or reviewed drafts of the paper, approved the final draft.

### Field Study Permissions
The following information was supplied relating to field study approvals (i.e., approving body and any reference numbers):
Soil samples were collected with the permission of the respective landowners:
1. Bert Schulze-Poll and Witiko Ludewig
Trantenrother Weg 25, 58455 Witten, Germany
2. Hubertus Schulte-Kellinghaus
Ekampsweg 1, 46244 Bottrop, Germany.

### Data Availability
The raw data are provided in the Supplemental Files.

## Supplemental Information

Supplemental information for this article can be found online at http://dx.doi.org/10.7717/peerj.5074#supplemental-information.

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
