# Peer review of "The effect of biochar amendments on phenanthrene sorption, desorption and mineralisation in different soils"

_PeerJ, doi:10.7717/peerj.5074_

## Round 0.1 · original submission · Major Revisions

The reviewers and I concur that this manuscript provides interesting insights on this topic, but several changes could improve the paper and make it more readable for a broader audience.

Please, pay particular attention to the remark made by reviewer #2 on the fact that "The general aim of the first two sections could be to provide a more detailed mechanistic discussion on biochar-soil interactions with a special focus on the differences between soils and biochars to subsequently help to explain the effect of soil type and biochar type on the sorption and degradation of phenanthrene."

·

Basic reporting

No comment

Experimental design

No comment

Validity of the findings

No comment

Additional comments

After reading, the manuscript entitled " The effect of biochar amendments on phenanthrene sorption, desorption and mineralisation in different soils ", I think the paper should be accepted as it. The topic is interesting for the scientific community and for the industry and it falls in the journal scope.
I think the work is quite novelty because of the biochar uses. The biochar is a form of charcoal obtained from wastes that it is being used as soil amendment. Despite of the thermic stabilization, the biochar is a material quite heterogeneous, strongly related to the type of the initial material used and pyrolysis conditions. Due to these characteristics, the studies under controlled conditions can provide information to understand and predict the biochar effect on soil properties and xenobiotic behaviour. The objective of this paper is to study the potential effect of biochar produced from three different feedstocks on phenanthrene (Phe) sorption/desorption properties and on its mineralisation. The experiment is performed according a classical design. This methodology is easily reproducible in laboratory conditions and allows to compare the results obtained from experiments made with other biochars and soils to evaluate the Phe sorption/desorption; but also, to compare results to evaluate the mobility of other organic xenobiotics with both soil and biochars studied in this experiment, but in different labs. The mobility of the organic xenobiotics from soils to the underground waters are controlled by the adsorption-desorption mechanism according to Freundlich isotherms. Finally, the use of 14C-labelled Phe to evaluate the mineralization provides precise information about the C mineralisation. In brief, the authors have presented a robust and classical model to describe the Phe mobilization in two soils and one sand amended with three different biochars. Both soils and biochars have been fully characterised according to routine lab protocols. The methodology is accurate to get the objectives proposed by the authors under lab conditions proposed. Soils, sand and biochars have not been sterilized, in principle, the role of the autogenous microorganism has not been quantified, the presence of controls can minimize the error. Results cannot be reliable to field conditions. However, the results can support the work hypothesis and the final conclusions in which as each biochar behaved slightly differently between soils, soil-biochar interactions should be considered in future research and caution is needed when extrapolating to other soils.

·

Basic reporting

The manuscript is written in clear English, at times the wording is slightly misleading and could be improved (e.g. lines 335-336, the use of “sorption capacity” when this parameter was not measured or calculated, for helpful insights into discussion of sorption data see e.g. https://doi.org/10.1016/j.watres.2017.04.014). Figures are relevant to the content and appropriately labeled. Unfortunately, during the introduction, as well as during the discussion of the results the authors neglect a portion of previous work on PAH sorption and degradation in the presence of biochar. For example, the fact that desorption from biochar is a rate limiting step for PAH degradation in biochar amended soils has previously been reported but was not discussed in the introduction and not used for the identification of knowledge gaps (e.g. http://dx.doi.org/10.1016/j.envpol.2013.06.026 and https://doi.org/10.1016/j.jhazmat.2017.11.010). Thus, it remains unclear to the reader what is already known, and how this study contributes to the field (see below for some suggestions). Accordingly, the sections at times appear to be less connected to the research aim than they could be and could be more focused on results relevant to the research aim(s).

Experimental design

The aim of the study is clearly stated, however, based on previous literature the authors could expect that phenanthrene sorption will depend on biochar type and soil properties. The data which were obtained fulfill high technical standards and surely have the potential to go beyond these expectable findings.

Thus, more attention should be given to the identification of existing knowledge gaps in the field, which could be useful to reshape the discussion and conclusion section of the manuscript. The results sections should be organized accordingly to better lead towards the answering of the previously identified knowledge gaps (currently for the first two sections this is not the case). The general aim of the first two sections could be to provide a more detailed mechanistic discussion on biochar-soil interactions with a special focus on the differences between soils and biochars to subsequently help to explain the effect of soil type and biochar type on the sorption and degradation of phenanthrene.
The discussion of Freundlich fits, especially differences in observed sorption nonlinearity (n) could be further elaborated in regards to sorption interactions and the effects of soil fractions on sorption to biochar. At the end of the sorption section the authors indicate that pH may have affected sorption of phenanthrene, but fail to explain why a neutral hydrophobic contaminant would sorb differently in dependence of soil pH.
The data presented in the phenanthrene mineralization section are very interesting and may offer more insight than is presented. For instance, the authors could look into the possible facilitation of abiotic phenanthrene transformation by biochar in the sand- biochar system to explain the different trend compared to soil systems. The section “integrative comments” is very helpful for the discussions of both sorption and mineralization measurements and the authors may consider to merge the two paragraphs with the respective previous sections. The concluding remarks could be sharpened in regards to key findings and how this study may be useful for further research in the field.

Validity of the findings

The data is robust, statistically sound and the key findings are valid with few remarks that need to be addressed (e.g. see comment on the effect of pH on phenanthrene sorption). The conclusion section should be sharpened towards the key findings and the meaning of these findings for the larger field.

Reviewer 3 ·

Basic reporting

This is an interesting paper that studied Phenanthrene sorption in biochar-amended soil. The results can be very valuable for people to understand the effect of biochar amendments on phenanthrene sorption, desorption and mineralisation in soil. The paper is well organized and largely written in proper English.

The authors wrote a long introduction, however, the significance and the innovation were not well presented. Great efforts should be done to improve it.

About Materials and methods, I have a question: What guideline or method did the authors used for performed Phenanthrene sorption experiment?
And I have a suggestion: Please, add the reference for the Phenanthrene mineralisation experiment.

Experimental design

This paper presents all the necessary requirements for the experimental design.

Validity of the findings

In think that this paper presents all the necessary requirements for the validity of the findings.

Additional comments

This is an interesting paper that studied Phenanthrene sorption in biochar-amended soil. The results can be very valuable for people to understand the effect of biochar amendments on phenanthrene sorption, desorption and mineralisation in soil

---

## Round 0.2 · Major Revisions

I appreciate the effort made by the authors to improve the manuscript. However, some of the points raised by the reviewers remain still open and need some additional changes. For example, reviewers ask for a deeper explanation about the relationship about low microbial activity and abiotic degradation. Several further remarks are reported in the annotated manuscript provided by reviewer 2.

·

Basic reporting

no comment

Experimental design

no comment

Validity of the findings

no comment

Additional comments

please see attachement

---

## Round 0.3 · accepted · Accept

I think that you have now replied to remarks provided by the reviewer and discussed properly the relationship of low microbial activity and abiotic degradation.